# Exploring Useful Teacher Roles for Sustainable Online Teaching in Higher Education Based on Machine Learning

**Yanni Shi** [1,2,*] 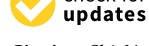 **and Fucheng Guo** [3]

1   Faculty of Artificial Intelligence in Education, Central China Normal University, Wuhan 430079, China
2   Department of Electronic Information, Henan University, Kaifeng 475001, China
3   Department of Artificial Intelligence, North China University of Water Resources and Electric Power, Zhengzhou 450045, China
*   Correspondence: syn@henu.edu.cn

**Abstract:** The COVID-19 emergency necessitated a rapid transition to online teaching by university lecturers. Hence, lecturers need to develop and reorganize their instructions and adjust their teaching roles and activities to the technological demands so as to further facilitate their continuous usage of technological systems after the crisis. Based on the behaviors of lecturers who utilized a particular teaching system—Rain Classroom—during school closure, this study aimed to predict their retention of online teaching beyond lockdown. Classical machine-learning classifiers were adopted to make predictions, most of which had an accuracy greater than 73%. Moreover, through a byproduct of these algorithms—feature scoring—we also aimed to determine the prime activities and roles that have strong relationships with lecturers' retention dispositions. The domain meaning of feature scoring was revealed based on a specific conceptualization of perceived usefulness and the TAM model, which further enlightened system devisers about strategies to improve technological quality. A coevolution mechanism was thus formed, both providing guidance for lecturers in changing their overt behaviors with respect to online teaching and supporting the customization of system functionalities, so as to foster the mutual adaption of teachers' pedagogies and artifact affordances. The findings, concerning useful teaching roles (namely, learning assessment, guiding technology usage, and learning support) and activities (such as in-class exercises, monitoring of students' attendance, formal testing, etc.), are corroborated by evidence from other reports in the literature.

**Keywords:** online teaching; teacher role; higher education; machine learning; Technology Acceptance Model; professional development; technology system evolution; distance education

## 1. Introduction

Online teaching and learning have been prominent research topics during the past three decades [1], with the value of this research greatly increased due to the COVID-19 pandemic which triggered school closures worldwide. The pandemic has presented not only challenges to teachers with respect to their preparations for moving online, but also an opportunity to introduce digital learning extensively into higher education [2]. Online teaching can serve differentiated/remedial instruction, as well as educational exploration, to expand educational capacities [3]; its sustainable development is therefore pivotal. However, without timely assistance for mandatory users to support their adaption to the relevant technologies, online experiences may be poor substitutes for face-to-face approaches and may have less of an influence on teachers' advancements into digital environments. Thus, the prediction—especially early prediction—of sustainable/unsustainable online engagement for individual teachers, as an effective measure and one of the objectives of the current study, can pave the way for prompt and precise support. Lecturers' fully online teaching during the crisis could evolve into a blended approach combining online and face-to-face learning after the pandemic, which is being encouraged by numerous universities in China.

Under the circumstances, sustainable online teaching, in the present study, refers to cases in which, when normal schooling resumes and offline learning is permitted, lecturers still spontaneously retain online teaching fully or partially for certain activities, as opposed to unsustainable cases in which lecturers flatly abandon the use of online methods. That is, sustainability means that teachers adopt online teaching.

As the suspension of classes during the emergency impelled the delivery of courses mediated by technology, with the learning processes supervised by teachers, the new technological environment entailed the renewal of teachers' roles and practices in ways different to those in traditional teaching contexts [4]. Accordingly, based on the prediction of an individual lecturer's continuance, the second objective of the present research was to analyze and identify the prime teacher roles and practices. These important activities in a particular technological context, as the strongest predictors of a teacher's retention of online teaching for all specific instructional tasks, can be further recommended with the relevant technology and pedagogical knowledge to educators to guide their overt behavioral changes. Teachers' roles are commonly defined as "functions and tasks involving teaching, usually well established by the educational institution, which indicate what teachers are allowed to do to carry out their educational activity" [5] (p. 5). Furthermore, tasks are broadly defined as "the actions carried out by individuals in turning inputs into outputs" [6] (p. 4). In this sense, the roles that teachers perform online can be reflected in and captured by the specific activities or behaviors they perform. Changing roles to suit online teaching essentially means that teachers need to adjust instructional behaviors and activities so as to be able to take on new sets of functions and tasks. Studies of online teacher roles are important, as they provide information about how online teachers might be trained and supported as well as factors that might affect the design of online learning environments [7]. However, previous contributions concerning online education have focused more on students' engagement and characteristics, and there is still a need for more research on teachers' characteristics [8]. Accordingly, it is worth exploring teachers' prime roles and activities in distance education, with those identified as being most correlated with sustainable engagement determining the prime practice scenarios at the current stage of introducing and boosting online teaching in higher education.

Renewing teacher roles as required for the educative use of ICT (information and communication technology) not only depends on the tasks and practices performed by university teachers, but also on the specificities of technological teaching environments [9]. The present study was conducted for an online teaching tool/artifact called Rain Classroom (RC), developed by XuetangX and Tsinghua University, which was widely utilized in higher education institutes in China during the COVID-19 pandemic. Lecturers' instructional activities, such as preparing and transmitting presentations, releasing bulletins, releasing exam papers, roll calls, video/audio deliveries, live broadcasts, monitoring student attendance and learning, and bullet chats, can all be supported by the functions of RC. These tasks can be further grouped into several categories to model distinct teacher roles.

In order to achieve the predictive aim of the current research, machine-learning techniques based on objective behavioral data were adopted due to the strong prediction power of the classification algorithms and the limitations of self-reported data. Additionally, machine-learning methods can make full use of the data resources automatically collected and stored in learning management systems [10]. Teachers' behavioral records indicate the online functions they typically undertake and thus link teacher roles with sustainable online teaching. A byproduct of the algorithms, feature scoring, can identify those activities that play important roles in prediction. Nonetheless, the importance of features for prediction in mathematical models is not necessarily equivalent to the importance of their corresponding features in the practical domain. Thus, searching for applicable theories in the domain of educational technology is crucial for yielding accurate elucidations and a deep understanding of teachers' online behaviors. Although AI could contribute copious state-of-the-art methods and techniques in the learning environment, there has been a critical gap between

what AI technologies can achieve and how they are actually implemented in authentic educational settings [11].

## 2. Rationale

### *2.1. Multidimensional Conceptualization of Perceived Usefulness in TAM*

Previous studies often applied the extended Technology Acceptance Model (TAM) to predict a technology system usage by conducting statistical analysis [12]. Within TAM, the two determinants of an individual's use intention are perceived usefulness (PU) and perceived ease of use (PEU), of which PU, defined as "the extent to which a person believes that using the system will enhance his or her job performance", is a fundamental driver of usage intentions [13] (p. 2). In the educational area, PU is also regarded as the strongest predictor of teachers' intention to use technology [14,15]. Two theoretic processes—social influence process and cognitive instrumental process—were incorporated to underlie the determinants of PU. This model, with general determinants of PU, referred to as TAM2, extended the original TAM and aimed to address the limitations in guidance for practitioners [13,16]. Complying with the cognitive instrumental process, individuals form a perceived usefulness judgment by cognitively comparing what a system is capable of achieving with what they need to achieve in their job. Four constructs—job relevance, output quality, result demonstrability, and PEU—capture this influence [16]. As university lecturers are pedagogically free and much empowered to decide their course organization, the cognitive instrumental process is considered as the dominant power on PU to explain teachers' voluntary usage after the emergency. However, the conceptualization of PU within TAM was operationalized at a general level, based on which most prior studies measured a unidimensional PU of an individual teacher, neglecting the specific goals of the teacher's work. Considering the complexity of teachers' job tasks, Scherer et al. [14] proposed an innovative perspective to conceptualize PU in aspects of multidimensional teaching purposes, namely, fostering students' interest and learning, collaboration and communication, and information retrieval. Nonetheless, this decomposition approach of PU focused on higher-level goals, concerning students' expected competencies rather than teachers' functions and practices, which created difficulty in determining what function category a certain teaching task should be grouped into. For example, based on Scherer's taxonomy, it is dilemmatic to decide the category of the activity of developing appropriate learning resources in a technological system. Hence, teachers' own actions may be more directly relevant for capturing specific PU towards the usage of a technological system.

The theoretical views described in the cognitive instrumental process are highly consistent with another important model—task technology fit (TTF)—in two aspects: the idea of judging the match between job task and technology, and the idea of decomposing constructs (PU and TTF) into detailed ones. TTF was defined as "the degree to which a technology assists an individual in performing his or her portfolio of tasks", and captured the fit "between system capabilities and user needs" [6] (p. 5). Hence, some researchers posited TTF as antecedents of core TAM to integrate both models for explaining technology use intention [17,18]. Nevertheless, TTF targets the impact of technology on job performance, highlighting the assessments of the characteristics of task and technology, whereas TAM points at usage intention and behavior, emphasizing individual perceptions. In addition, when measuring TTF, the original TTF study developed an instrument with eight components stemming from the demands of the information system [6], while other studies we reviewed, such as [19], often relied upon a perceived task–technology fit based on self-reported data. In contrast, within TAM2, four definite constructs (job relevance, output quality, result demonstrability, and PEU) were conceptualized to capture the users' mentally matching process in more detailed dimensions. As such, the TAM2 model, extended by PU's determinants, is more capable of underpinning an in-depth analysis of why some particular activities and roles are perceived to be more useful. Consequently, considering the predictive target of continuous use behavior and attribution analysis, we

adopted TAM2 with specific PU as a basis to gain insights into lecturers' online teaching practices and the domain meaning of feature importance scores.

*2.2. Teacher Roles in Online Teaching*

As described earlier, online teacher roles imply different functions to ensure teaching presence in a digital learning environment, which groups and clusters the specific activities well established by educational institutions. In other words, an online activity, belonging to a certain category of teacher roles, is a lecturer's execution of a set of skills or a teaching task. For this classification problem regarding the instructional practices, researchers have created diverse taxonomies and models specifying teaching roles through observing and analyzing teachers' experiences in a virtual environment. However, these studies showed diversity in context and definition, and the prioritization of the roles and competencies also varied across the literature, depending upon the context where online teaching took place [7]. For example, Garrison et al. [20] proposed the Community of Inquiry (COI) framework, and contended that successful online education requires three crucial prerequisites—cognitive presence, social presence, and teaching presence. Thereof, teaching presence is composed of three discrete functions: design and administration, facilitating discourse, and direct instruction [21]. In another well-known study published by IBSTPI, the work on defining teachers' professional competencies for online teaching started by identifying the specific roles and tasks. Five domains of teacher performances were linked to their functions: professional foundations, planning and preparation, instructional methods and strategies, assessment and evaluation, and management [22]. Badia et al. [5] (p. 5) integrated previous significant contributions to obtain a role model with five types of teachers' roles in online teaching: "(1) instructional design, which includes teaching activities related to educational planning; (2) managing the learning activity, which refers to the organization of the learning tasks during the course; (3) learning assessment, which refers to how to monitor students' learning; (4) managing social interactions, which includes activities that promote appropriate social interaction; and (5) design and educational use of technology, which refers to teachers' actions to guide the appropriate use of technology". Subsequently, based on this role model, Badia et al. built a Likert-type scale to measure teachers' preferred roles through their evaluation of the importance of the relevant tasks. In other words, this measurement captured teachers' perceived importance of online roles based on self-reported data, rather than the actual roles in their day-to-day teaching behaviors which may be restricted by factors from the environment such as technology settings. In order to identify the importance of teachers' diverse online functions, Bawane and Spector [23] developed a list of comprehensive roles, and investigated the prioritization of these roles based on expert opinion. Among those eight roles, pedagogical role received the highest priority, followed by professional, evaluator, social facilitator, technologist, advisor, administrator, and researcher. The present study pursued the useful teacher roles that should not only conform to sustainable online practices, but also adapt to educational surroundings. Considering the specific affordances offered by RC, we mainly incorporated the role model given by Badia et al. [5], and modified it slightly to discern and fit lecturers' activities in the RC system. The model contains five roles: instructional design and presenting content, managing social interaction, learning assessment, guiding the use of technology, and learning support. Each category further consisted of specific tasks and activities, as listed in Table 1.

**Table 1.** Model of teacher roles.

| Teacher Roles | Tasks and Activities |
| --- | --- |
| R1: Instructional design and presenting content | Design instructional strategies<br>Develop appropriate learning resources<br>Offer specific ideas/expert and scholarly knowledge<br>Demonstrate effective presentation |
| R2: Managing social interaction | Promotion of relationships of trust and mutual commitment among students<br>Enhancement of cordial and warm relations between teacher and students<br>Resolution of group conflicts among students<br>Facilitation of personal or professional knowledge sharing among students |
| R3: Learning assessment | Correction of students' misunderstanding of content<br>Providing students with information about assessment (grades, correct answers, and/or evaluation criteria)<br>Resolution of questions from students about the content<br>Monitoring and evaluation of students' individual and group activities |
| R4: Learning support | Guidance and regulation of students' individual study processes<br>Control and monitoring of students' learning pace and learning periods<br>Guidance, monitoring, and evaluation of students' participation in learning activities |
| R5: Guiding the use of technology | Guidance of students in the use of the virtual learning environment<br>Regulation of an appropriate use of technology by students<br>Design of certain technological tools for learning<br>Decision to integrate new technological tools into the existing virtual environment |

### 2.3. Machine Learning Techniques for Prediction of E-learning

Along with the maturity of machine learning techniques, some studies adopted this way or a hybrid approach, namely, a classical SEM and machine learning method to analyze predictive models towards the use behavior of learning systems. For example, on the foundation of students' practical roles of information management (i.e., retrieval, storage, sharing, and application), Arpaci [12] collected self-reported data from undergraduates to predict their behavioral intentions towards educational usage of mobile cloud computing. A complementary approach of machine learning algorithms alongside a classical SEM was adopted with predictive accuracy exceeding 72% in most classifiers. Similarly, Akour et al. [24] gathered self-reported data from a survey of university students in the United Arab Emirates to validate a model of predicting mobile learning platforms usage during the COVID-19 pandemic period. Among several machine learning techniques, decision tree J48 performed best in predictive accuracy.

The above-mentioned studies essentially pertain to analytic examination, which is descriptive rather than predictive for future targets [25]. The current research problem falls more into a homogeneous issue—student dropout prediction (SDP) in the field of learning analytics. According to the survey [25], the majority of research addressing SDP relied on off-the-shelf machine learning algorithms, among which the widely adopted techniques encompassed decision tree, support vector machines, Bayes classification, neural networks, logistic regression, etc. Prior contributions organized the related literature and concluded that two types of data were exploited as attributes to predict student retention, namely, time-invariant attributes (such as demography and educational level) and time-variant features (such as clickstreams and forum interventions) [25,26]. For the activities with temporal dimension, studies extensively flattened them into summary statistics (derived from all phases) for a plain modeling schema, while some studies treated them as time series [25]. A byproduct of the algorithms, variable importance score, was typically used to indicate the significant predictor. For instance, Mendez et al. [27] employed the techniques of classification trees and random forest to predict student persistence in science and engineering majors and to obtain associated factors. Hu et al. [28] used decision trees (CART and C4.5) and ensembles (AdaBoost) to construct an early warning system, and flattened the activities into plain data for feature importance analysis. They found that time-dependent variables extracted from learning management systems are the critical factors for online learning. Qiu et al. [29] proposed an integrated framework with the function of feature selection to predict student dropout in MOOCs, which scored all features using the methods of mutual information, random forest, and recursive feature elimination. Gray and Perkins [30] applied Nearest Neighbor (1-NN) classifier and decision tree C4.5 (J48)

to forecast risk students and to obtain relevant features. Panagiotakopoulos et al. [31] employed nine algorithms, including Ridge, gradient boosting classifiers, and logistic regression, to predict students' dropout in MOOC at an early stage, and recorded the respective feature importance scores of online practices by different classifiers. Built on these validated methods, the experiment in this research would trial prediction targeting teacher retention of online teaching, rather than student retention of online learning, using both types of behavioral data (flattened data and time series), and would explore the relevant features through unitizing the importance scores obtained from multiple classifiers. Despite a plethora of high-performing studies on student dropout prediction, we cannot necessarily believe the same satisfactory performances for teachers. Going further, we also attempt to uncover the domain meaning of feature scoring, explore behavioral styles for sustainable online teaching, and support both teachers and artifact devisers.

### 2.4. Building Predictive and Descriptive Models for Present Study

Gaudioso et al. [32] declared that there exist two different types of models supporting adaptive learning systems. On the one hand, predictive models require a particular target of interest in advance and help the teacher to detect or anticipate problematic situations in the students' learning process; On the other hand, descriptive models extract patterns of common student behaviors and allow teachers to analyze what has happened in a learning situation. Accordingly, the current study has both an interesting target to predict and a descriptive modeling flavor to analyze lecturers' online behaviors and roles. Thus, unlike those purely predictive modeling tasks which pursue high prediction accuracy, this research has considerations of comprehensibility and interpretation. Grounded in the aforementioned rationale, we build a combined model to forecast teachers' retention in online teaching, as well as analyze teaching roles based on their specific activities performed online, which is illustrated in Figure 1. Drawing on the role model from [5], five teachers' roles (i.e., instructional design and presenting content, managing social interaction, learning assessment, learning support, guiding the use of technology) portray lecturers' job functions in the RC system and capture behavioral patterns for sustainable online teaching. The attributes for prediction are the statistics of fine granularity activity records of lecturers. Classical machine learning algorithms are employed to fit the behavioral data and to yield feature importance scores. Based on the feature scores, prime teacher activities and roles can be derived using an ensemble strategy, so as to gain patterns of sustainable online teaching.

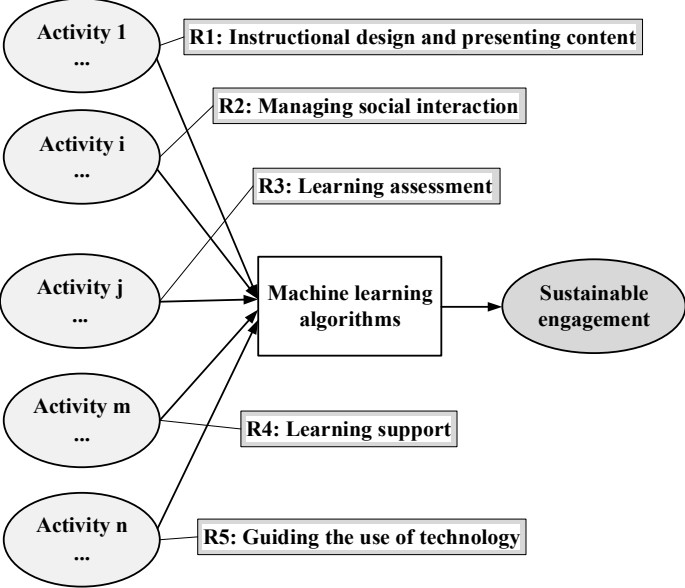

**Figure 1.** The predictive model.

## 3. Materials and Methods

### 3.1. Context and Data Description

This research is conducted at a full-time comprehensive university located in central China. The university has been promoting education informatization for several years and is equipped with sound facilitation conditions, for instance, full coverage of campus network, digital learning platforms such as RC prepared in most physical classrooms, training videos available on intranet, and a local support center staffed by specialists. Since the outbreak of COVID-19, university teachers have occasionally been required to transition online in response to the sometimes-severe pandemic. When schooling returns to normal after a crisis situation, lecturers are permitted to freely select their teaching approaches: online in almost the whole teaching process with platform installed and utilized in the classroom, online partially for enjoyable activities, or offline completely. The first two cases are classified as sustainable engagement, as mentioned earlier.

The target population in this study are the lecturers who were responsible for the undergraduate teaching tasks during the autumn semester in 2021. There was a total of 18 academic weeks and more than 5500 undergraduate classes launched in that term. However, due to another round of pandemic risk, the target university, originally scheduled to start the new semester in September, had to close until the crisis eased in the third academic week. Lecturers were forced to deliver courses fully online during the first three weeks. Diverse educational or commercial learning systems were available, and teachers could choose their favorite one for course delivery. The number of teachers who used RC during the first three weeks was 1548, accounting for the largest proportion compared with other technological platforms. At the beginning of week 4, lecturers returned to normal teaching but were permitted to use any of the three teaching approaches mentioned above. At this time, the number of teachers who voluntarily retained platform usage plummeted to 458. The weekly number of users then remained moderately steady until week 16. The last two weeks were reserved for university-wide final examinations, during which teachers did not present lessons. The detailed weekly number of RC users in that semester are shown in Figure 2. The number in the first week was less than that in the second/third week, probably because some teachers were unable to enter their large online classes due to server overload in the first two days and temporarily transferred to other commercial systems for teaching.

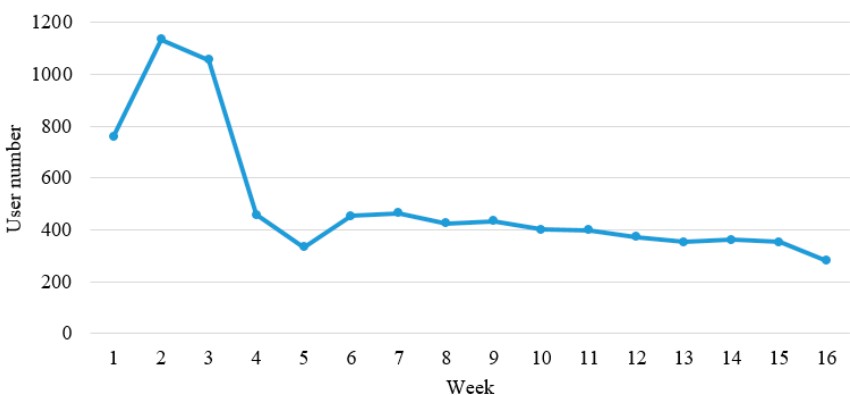

**Figure 2.** Weekly number of Rain Classroom users.

For the dataset that we analyzed, teachers' personally identifiable information (e.g., name, address, employee ID, and demographic information) and course information (such as course name and ID) that can identify an individual when combined with other data were all anonymized to avoid ethical and privacy issues. Only records concerning teachers' professional behaviors, which are open in their organization and can be observed by other members and management departments for evaluation and intervention, were reserved for our research. This statistical data of the activity records was provided by the RC platform rather than extracted from raw logs by us. A university is allowed to download statistics

pertinent to its own staff, so as to learn how online teaching is progressing. Lecturers can see data generated by themselves or the students in their classes conveniently on the RC platform to assess teaching and learning performances. Additionally, in order to analyze and understand online activities in depth, a survey of teachers was conducted to supply more information on which functions and activities lecturers perceived to be useful/useless in the RC system and the reasons why. As this research progressed, we interviewed lecturers several times, with the total number of respondents more than 20. Using content analysis and narrative analysis, their responses were grouped into several issues, including decisions regarding adoption or retention of online teaching, useful/useless functions and activities in RC, and the advantages and disadvantages of a specific activity. All information was then aggregated to analyze the research results in the Discussion section.

### 3.2. Problem Statement and Feature Extraction

The prediction task of the present study is addressed in the way that copious studies solved SDP. Since each course was launched based on weeks, a time window with a weekly unit for time slice was employed to illustrate the predictive target and features (see Figure 3). The time axis started at the beginning of the semester when course content was delivered wholly online. Features were extracted from the statistics of teachers' activities related to mandatory use of a technological system during the historical period of the first three weeks. Being a supervised binary classification task, the target had two classes: one was the sustainable case in which a teacher would still perform online activities in the future period, coded as 1; the other was the unsustainable case in which a teacher would not have any online activities in the future period, coded as 0. Nevertheless, the length of the future period for prediction is difficult to decide. If it is too long, then the prediction will be late and the system will be unable to offer prompt intervention; if it is too short, then some engagement records will be excluded and the classification will be inaccurate. Drawing upon the typical way that the problem of SDP was addressed, a duration of four weeks (a month) was determined to judge the targeted class. Thus, the prediction problem in this study can be defined as predicting whether a teacher will retain specific activities in RC in the next consecutive four weeks based on his/her mandatory use behaviors during the historical first three weeks. The number of sustainable cases among the 1548 RC users was 678, accounting for 43.8%.

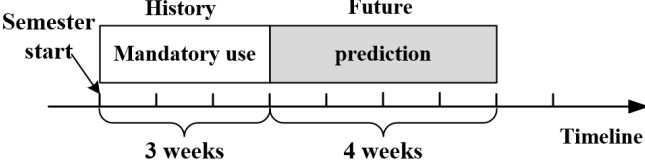

**Figure 3.** Prediction problem definition based on a time window.

The features of prediction were the statistics of lecturers' online activity records, which were calculated through aggregation functions (e.g., count or percentage). Each feature name had two parts. The front part indicated the statistical method adopted for calculation, while the rear part referred to a particular online activity lecturers performed. There was a total of 14 features indicating the specific activities to model teachers' presence in the virtual environment, of which the statistics of 13 features were directly offered by the platform while the last one was the previous derivation. Table 2 lists these features and their brief descriptions, together with the matching teaching role (see Table 1). Since these behavioral features had temporal dimensions, we employed two common methods to process the time property. One method was flattening data as summary statistics across the whole period, such as [28,32]; the other method was assembling the activities by week as separate variables (i.e., time series), such as [33]. The dataset generated with the former method was labeled as T3W, denoting the summary statistics for the first three weeks; the dataset generated with the latter method was labeled as T111W, denoting the weekly statistics within the first three weeks. In this way, T111W actually included a set of new features that

were time series made up of 42 (14*3) statistics of activities. Since the three-week duration was a short span, we inferred that the time dimension might have little influence on the predictive performances of the machine learning models. Furthermore, the plain data of summary statistics were more reasonable and feasible to directly retrieve the important activities. Despite these considerations, we tested both datasets and the comparative results confirmed our speculation, as shown in the next section. The results yielded on T111W were then abandoned and the follow-up analysis of teachers' roles merely depended on the results of T3W.

**Table 2.** Feature set related to activities.

| Feature Name | Description | Teacher Role |
|---|---|---|
| Avg_AttendRate | Monitor the average rate of students attending online classes over a certain time period | R4 |
| Num_Material | The number of course materials (mainly PPTs) a lecturer uploads over a certain time period | R1 |
| Avg_ReadRate | Monitor the average rate of students reading the provided materials over a certain time period | R4 |
| Num_Test | The number of formal test papers a lecturer uploads for student assessment over a certain time period | R3 |
| Avg_CompleteRate | Monitor the average rate of students completing uploaded papers over a certain time period | R4 |
| Num_Bulletin | The number of bulletins a lecturer issues to inform students of learning arrangements over a certain time period | R4 |
| Avg_ViewRate | Monitor the average rate of students viewing the issued bulletins over a certain time period | R4 |
| Num_Exercise | The number of exercises presented by a teacher to test and correct his/her student's learning during a live broadcast in class over a certain time period | R3 |
| Num_Writing | The number of pieces of writing submitted by a lecturer or his/her students to an e-learning wall for sharing ideas during a live broadcast in class over a certain time period | R2 |
| Num_Bulletchat | The number of pieces of bullet chats between a lecturer and his/her students during a live broadcast in class over a certain time period | R2 |
| Num_RedEnvelope | The number of times a lecturer rewards his/her students performing well with money during a live broadcast in class over a certain time period | R4 |
| Num_PuzzledPPT | Monitor the number of incomprehensible PPT slides students reported over a certain time period | R3 |
| Num_Call | The number of roll calls a lecturer made randomly to interact with a particular student during a live broadcast in class over a certain time period | R2 |
| Num_Type | The number of types of activities a lecturer performed on the platform over a certain time period | R5 |

### 3.3. Classifiers and Feature Scoring

In order to select the appropriate machine learning algorithms, we considered several relevant issues, including the size of the gathered dataset (1548 instances), diversity of mathematical meanings of feature scores, and the most adopted and well-performed classifiers. As such, deep learning algorithms, generally using models with complex structures and performing well in prediction, were excluded due to overfitting problems and the weakness in feature selection. Because different models have totally different opinions on classification and feature importance, nine classifiers were finally applied to predict the dichotomous outcome. These techniques and the specific algorithms, implemented in Jupyter Notebook workbench with Scikit-learn library, are listed below:

1. Logistic Regression (LR) with L2 regularization [34];
2. Ridge classifier (Ridge) [35];
3. SVM, Support Vector Classification (SVC) with linear kernel [36];
4. Decision tree, Classification and Regression Tree (CART) algorithm [37];
5. Ensemble method, Gradient Boosted Decision Trees (GBDT) [38];
6. Ensemble method, AdaBoost boosting algorithm (AdaBoost) [39];
7. Ensemble method, Random Forest (RF) classifier [40];
8. Gaussian Naive Bayes (GNB) algorithm [41];
9. Neural network, Multi-Layer Perceptron (MLP) algorithm [42].

Thereof, the linear models added L2 regularization to penalize some coefficients for confronting the overfitting problem, which in turn reduced the effect of multicollinearity. Before training the above nine models, both datasets (T3W and T111W) were standardized and variance inflation factors (VIFs) were computed to diagnose the collinearity of features for assuring accurate estimates of how much each attribute could affect the target variable. A byproduct of the algorithm (i.e., feature importance scoring) was then deployed to exhibit the contribution of each predictor to prediction. Specifically, within linear models, coefficients are regarded as feature importance. Within tree-based techniques, a feature such as a decision node in a tree has a relative rank (i.e., depth) that can be used to assess the relative importance of the predictability on the target variable [43]. These models offer a property called "feature importance" to show the feature scores in Scikit-learn. Within the MLP and GNB algorithms, permutation feature importance was calculated to score and sequence features for these opaque estimators [40].

Since feature importance scores derived from different classifiers might significantly vary from each other, an ensemble strategy was adopted to combine the ideas over all comparison models (namely, LR, Ridge, SVC, CART, GBDT, AdaBoost, RF, GNB, and MLP). Thereof, LR, Ridge, and SVC produce directional weight coefficients with sign (+/−) indicating classification (1/0). Thus, pursuing a comprehensive and thorough understanding of lecturers' online activities, we considered the situation of negative influence and dealt with feature scores in two stages. In the first stage, all feature scores were treated as non-directional, and values only gave an indication of the magnitude of relevance to data separation. For this case, two detailed steps were taken: firstly, normalizing absolute feature scores between 0 and 1 by different classifiers; secondly, averaging the normalized scores of the same feature from all classifiers. Building on the first stage, the second stage focused on features with a negative score yielded by LR, Ridge, and SVC. Considering domain interpretations of features with negative scores, especially the negative high value obtained in the first stage, we could gain more insights into teachers' online behaviors. Furthermore, in order to prioritize different teachers' roles in facilitating their sustainable online teaching, scores of the features belonging to the same teacher role were averaged to generate a unique value for ranking them.

## 4. Results

### 4.1. Comparison of Prediction Performances

The experiment performed on dataset T3W is tagged as Track1, with its companion experiment based on dataset T111W tagged as Track2. Each dataset was divided into two parts—80% of data for training and the remaining 20% for testing—based on a standard train–test data-splitting approach [33]. Hold-out validation was adopted to evaluate the performances of classifiers, except linear models that used 10-fold cross-validation for searching optimal regularization parameter values and testing the stability of coefficients. A mixture of well-known metrics, i.e., accuracy, weighted precision, weighted recall, weighted F1-score, and area under the ROC curve (AUC), provided an intuition of the overall performances of a model. Due to the imbalanced dataset for target categories, we preferred the F-measure [44] and AUC [33] for references.

The prediction results of nine comparison models performed in Track1 and Track2 are shown in Table 3. Looking at the overall metrics, all classifiers were greater than 70%.

Considering the short span (only three weeks) of lecturers' mandatory usage, these prediction results manifest a strong relationship between teachers' retention of online teaching and their behavioral patterns. Among these models, MLP demonstrates the best predictive ability because it has the highest values in all metrics based on datasets with/without a time dimension. Apart from MLP, classifiers, including LR, SVC, and AdaBoost, also show higher metrics values in both Track1 and Track2, although the differences in overall performances among all nine models are not large. In Track1, Ridge is slightly weaker in prediction performance than the other classifiers, as reflected in low metrics values for accuracy (0.71), F1-score (0.70), and AUC (0.72), yet it is still a useful model due to its strength in dealing with the collinearity among attributes. Comparing Track1 and Track2, we can see that an additional temporal dimension does not contribute much to a more excellent forecasting performance. Most models in Track2 just improve accuracy slightly, while a few classifiers, such as GBDT and RF, perform even worse according to several metrics. These results confirm our presumption that time series with a temporal feature in this research impacts little on the enhancement of prediction ability.

**Table 3.** Performance results of applied classifiers in Track1 and Track2.

| Classifier | Track1 | | | | | Track2 | | | | |
|---|---|---|---|---|---|---|---|---|---|---|
| | Accuracy | Precision | Recall | F1-Score | AUC | Accuracy | Precision | Recall | F1-Score | AUC |
| LR | 0.75 | 0.75 | 0.75 | 0.74 | 0.74 | 0.76 | 0.76 | 0.76 | 0.75 | 0.75 |
| Ridge | 0.71 | 0.70 | 0.71 | 0.70 | 0.72 | 0.75 | 0.74 | 0.75 | 0.74 | 0.74 |
| SVC | 0.75 | 0.75 | 0.75 | 0.74 | 0.75 | 0.77 | 0.77 | 0.77 | 0.76 | 0.76 |
| CART | 0.75 | 0.77 | 0.75 | 0.73 | 0.72 | 0.75 | 0.76 | 0.75 | 0.74 | 0.71 |
| GBDT | 0.73 | 0.73 | 0.73 | 0.73 | 0.75 | 0.72 | 0.71 | 0.71 | 0.71 | 0.76 |
| AdaBoost | 0.75 | 0.75 | 0.75 | 0.75 | 0.77 | 0.75 | 0.75 | 0.75 | 0.75 | 0.77 |
| RF | 0.72 | 0.73 | 0.72 | 0.72 | 0.75 | 0.71 | 0.72 | 0.71 | 0.71 | 0.78 |
| GNB | 0.74 | 076 | 0.74 | 0.71 | 0.73 | 0.76 | 0.78 | 0.76 | 0.74 | 0.74 |
| MLP | 0.77 | 0.77 | 0.77 | 0.76 | 0.77 | 0.77 | 0.77 | 0.77 | 0.76 | 0.79 |

### *4.2. Detecting Collinearity*

As mentioned earlier, weight coefficients in linear models would be indicators of feature importance. However, the existence of collinearity in the LR model inflates the variances of the parameter estimates, and consequently leads to incorrect inferences about relationships between explanatory and response variables [45]. Thus, variance inflation factors (VIF) were calculated on both datasets (T3W and T111W) to detect collinearity among independent attributes. As a rule of thumb, a VIF value exceeding 10 indicates a collinearity problem. Table 4 lists the detailed VIFs of T3W. The highest two values are 7.32 for Num_Test, and 7.05 for Avg_CompleteRate. The remaining VIFs are all less than 5. These results show that T3W is within tolerance to collinear relationships in the LR model. As for T111W, the 42 VIF values are not displayed in detail due to space limitations of this article. There are four VIF values greater than 10 (specifically, 17.38, 16.57, 11.88, and 11.67), while the remaining values are less than 7. This indicates strong collinearity, which will cause an inexact ranking result of feature importance as a result of unstable and inaccurate estimation of weight coefficients. Because of the collinearity problem and insignificant contributions to prediction, Track2 is terminated at this moment and the subsequent analysis is made on dataset T3W.

**Table 4.** VIFs of features in dataset T3W.

| Feature | VIF | Feature | VIF |
|---|---|---|---|
| Avg_AttendRate | 4.04 | Num_Exercise | 1.54 |
| Num_Material | 2.09 | Num_Writing | 1.06 |
| Avg_ReadRate | 2.08 | Num_Bulletchat | 1.35 |
| Num_Test | 7.32 | Num_RedEnvelope | 1.04 |
| Avg_CompleteRate | 7.05 | Num_PuzzledPPT | 1.37 |
| Num_Bulletin | 2.49 | Num_Call | 1.07 |
| Avg_ViewRate | 3.14 | Num_Type | 4.76 |

### 4.3. Feature Scores and Ranking

Following the processing steps described in the previous section, the normalized feature scores from all nine models are displayed and compared in a heatmap (see Figure 4). For each attribute, the average score is plotted in Figure 5, alongside the rankings of 14 features in descending order. As we see in Figure 4, colors in different models for the same feature show significant discrepancy. Different classifiers indeed generate obviously discordant importance scores due to their separate mathematical emphasis. Most features exhibit this characteristic, except Num_Exercise and Num_RedEnvelope, which have the highest and lowest score, respectively, and appear to be moderately consistent in all classifiers. For models—namely, LR, Ridge, and SVC—that yield feature scores with the sign representing a directional effect on the target variable, the negative weights are labeled using "-" in the cell, so that we can further discuss their domain meaning and guide practices. As shown in the heatmap, these three models express a unanimous opinion that Num_Call and Num_Material are negatively correlated to targeted 1 class (sustainable engagement).

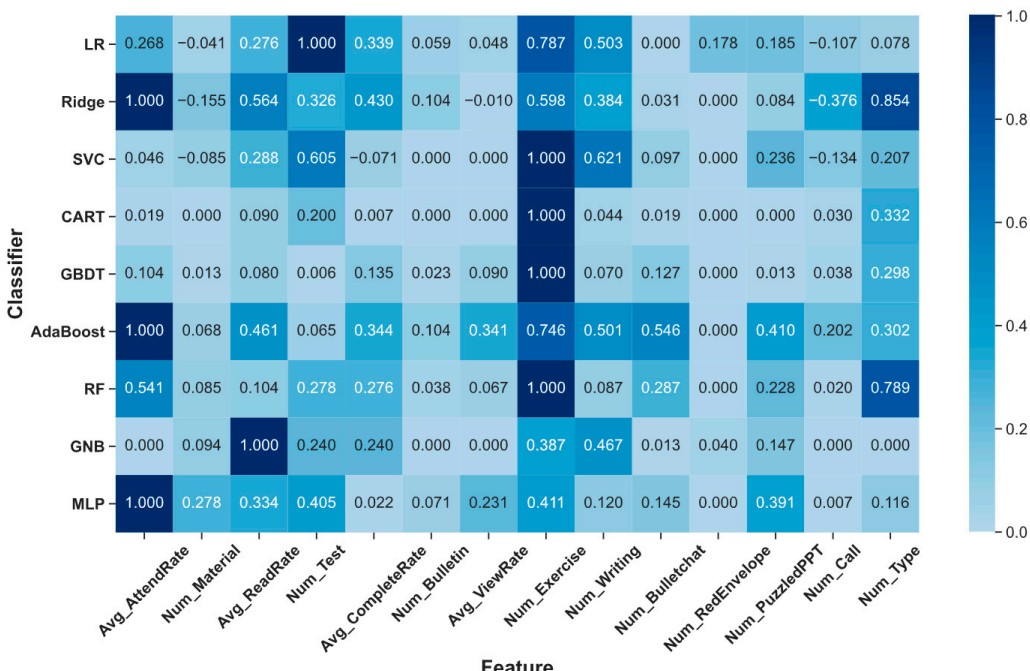

**Figure 4.** Heatmap of feature importance for different classifiers.

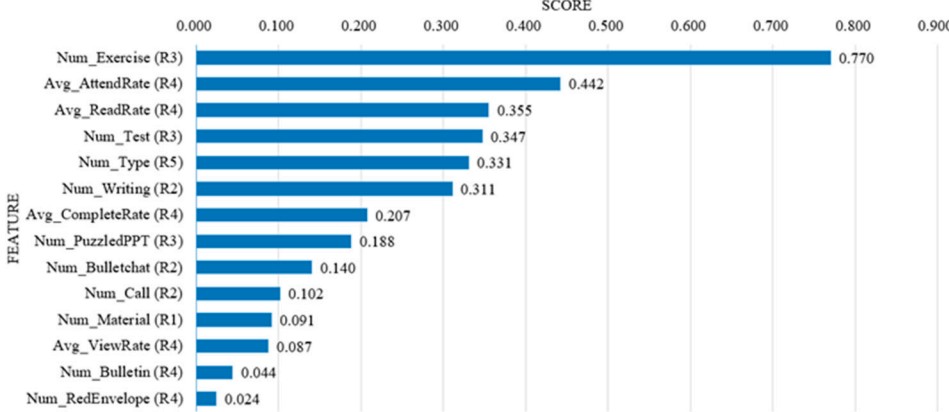

**Figure 5.** Feature ranking in descending order.

Based on the combined importance scores, Figure 5 presents the ranking of features in descending order, as well as their corresponding teacher roles. Num_Exercise recorded the highest score (0.770), outnumbering the score (0.442) of the second feature Avg_AttendRate. Num_Exercise records the number of in-class quizzes that teachers perform to assess and correct students' learning acquisition, belonging to learning assessment (R3), whereas Avg_AttendRate involves the activity of monitoring students' attendance, related to learning support (R4). The following four features have almost equal importance values, pertaining to the roles of R4, R3, R5, and R2, respectively. The subsequent six features recorded lower scores, of which Num_Call, in 10th position, and Num_Material, in 11th position, are both negatively associated with sustainable online teaching. The remaining two features, Num_Bulletin (0.044) and Num_RedEnvelope (0.024), had marks much lower than the others.

By averaging the scores of features that belong to the same role, the teacher roles are sequenced in Figure 6 in descending order: learning assessment (R3), guiding the use of technology (R5), learning support (R4), managing social interaction (R2), and instructional design and presenting content (R1). It should be noted that learning support (R4) contains six relevant activities, three of which recorded the lowest scores in the feature ranking and thus largely lower the rank of R4. These outcomes present the important teacher roles or specific activities that are more correlated to lecturers' sustainable engagement, and therefore can reveal the different behavioral patterns of two types of teachers in an online environment.

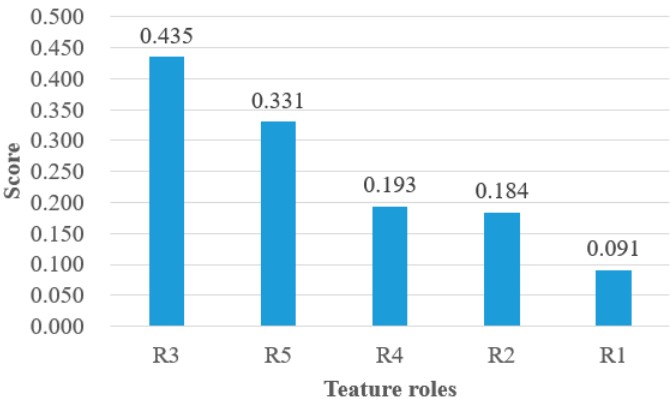

**Figure 6.** Teacher roles ranking.

## 5. Discussion

Since the ranking results of teacher roles and activities are obtained through a byproduct of machine learning methods, our analysis of the results should also surround the byproduct—feature score. For linear models, it has four cases: high positive value, high negative value, low positive value, and low negative value. For tree-related models and MLP, it has no directional influence on the target, but a high or low value. Underpinned by the rationale elucidated in the second section, feature importance scores can be understood at two levels. At the mathematical model level, a feature score shows the strength of the covariance relationship between the feature and targeted variable. Feature-related activities with higher scores have a stronger covariance with sustainable online teaching than those with lower scores. In other words, the activities with high scores are the strong predictors forecasting lecturers' retention. For linear models, such as LR, feature scores are equivalent to feature coefficients or weights, which inherently have a directionality of covariance (+/−). As a result, the more teachers practice the activities with high positive/negative values, the more they are inclined/declined to continue voluntarily. However, it should be noted that the relationship between feature-related activities and teachers' retention is not causal. That is, activities with high scores are not the significant determinants that can explain sustainable online teaching, but are the powerful predictors or indicators on sustainability. Actually, lecturers' activities and their retention are both extracted from

the same behavior database, and this research is predicting lecturers' future behaviors based on their historical behaviors. In order to determine why certain activities can possess more prediction power than others, we should logically resort to interpretable domain theories to explain the domain sense of feature scores. Because all specific activities are homogeneous with their respective feature scores having the same domain meaning, a model concentrating on one dominant factor is applicable to uncover this meaning. As such, the single dominant factor sheds light on the discrepancy of feature scores for different activities. As expounded earlier, PU specific to detailed job tasks within TAM2 is a perfect fit for intelligible domain meaning. Accordingly, a feature score at the level of domain sense means the extent to which lecturers perceive to be useful by performing the feature-related activity with a given teaching system. The more teachers practice the activities with high positive scores, the more useful perceptions they can accumulate to retain online engagement.

Further, in order to gain insights into why different activities would obtain varied importance values, we zoom into PU's determinants to use second-order factors explaining the discrepancy of PU for specific tasks. As described in the second section, PU is judged through two theoretical processes: cognitive instrumental process and social influence process. In the former, teachers form PU judgement by cognitively comparing what a system is capable of achieving with what they need to achieve in their job, so that factors relevant to job tasks and system characteristics are covered in this process. As job tasks in fact refer to teaching activities, through them, we can just tap into each feature to analyze why it is perceived to be useful or useless (indicated by the size of the feature score). Although factors involving the social influence process (such as subjective norm and image) also influence teachers' PU, they are indirect to job tasks and have a general impact on every specific activity, and thus are ineffective in explaining the scoring discrepancy of different activities. Consequently, the constructs (namely, job relevance, output quality, result demonstrability, and perceived ease of use) that convey the cognitive instrumental process underpin an analysis of the detailed reasons why lecturers would perceive such an extent of usefulness when accomplishing a particular job task with the given technology system.

In the context of this research, specific PU means the extent to which teachers perceive usefulness by practicing a particular activity in the RC system. Job relevance catches whether a particular activity matches a job task; output quality represents how well an activity could be performed via RC; result demonstrability indicates the discernibility of covariance between performing an activity via RC and positive results; PEU captures lecturers' efforts to perform an activity using RC. Accordingly, a teacher would retain online teaching, because s/he has perceived online implementation to be useful enough for her/his job goals. The more a teacher engaged in the activities with large positive feature scores during compulsory usage, the more usefulness s/he would perceive to continue the online approach; however, more engagement in activities with small or negative scores would have an insignificant effect on or even beget uselessness accumulation for lecturers' retention. Thus, the activities with high scores manifest a behavioral pattern of sustainable online teaching. A useful activity implies that it is not only an important teaching task, but also has high output quality, high result demonstrability, and high PEU via the technology system. Conversely, an activity appears useless because it is perceived as an unimportant job task, or has low output quality, or low result demonstrability, or low PEU, or a combination of these aspects.

Underpinned by the domain meaning of feature scores, the results yielded by the current experiment can be analyzed to understand behavioral patterns for online teaching. Based on the rankings of roles and activities, learning assessment is the most useful teacher role on the RC platform, including three activities: in-class exercises and quizzes, formal testing, and puzzled PPTs. These activities all recorded relatively high scores and fulfill a common function—assessing students' learning outcomes and correcting misconceptions—which suggests that lecturers should pay more attention to students' learning effectiveness.

The second useful role, guiding the use of technology, contains only one feature—teacher-led various technologies usage—derived from counting the number of types of system functions that a lecturer has used throughout a period. Diverse technologies usage can advance teachers' software skills and facilitate their adoption of the technology platform. The more and varied technological experience teacher candidates have, the more and varied use they could imagine [46]. Li et al. [47] also claimed that the number of learning sessions designed by teachers (referring to the diversity of types of teaching activities) is significantly related to teaching effect, which further positively correlates to teachers' support attitudes to online instruction during the pandemic. Although the third role, learning support, shows only slight disparity in its score with the fourth role, managing social interaction, it encompasses six activities with the three lowest scores, decreasing its ranking position. Among six activities, three useful ones are monitoring students' attendance at synchronous lectures, reading of learning materials, and completion of uploaded papers. The remaining activities are perceived to be useless due to distinct weaknesses. Specifically, the activity with the lowest score is rewarding outstanding students with money (red envelope). Although lecturers can perform this activity conveniently and safely in the RC system with the help of the online payment function of WeChat, it should be regarded as an unimportant job task because no more than 10 teachers out of 1548 RC users performed this activity. As for activities—issuing bulletins and monitoring students' view of them—lecturers reported that they often employed a familiar social media app, WeChat, instead of RC to inform students about learning assignments, which causes poor discernibility for results demonstrability. Given the three lowest values, the role of supporting learning to guide and monitor learners' study processes should have been more important for online teaching tasks, just as Baran et al. [48] argued that increasing the teacher presence for monitoring students' learning is one of the greatest pedagogical changes when transforming to an online environment.

Managing social interaction is always considered to be important for online learning, as the social constructivism theory revealed, yet it was not perceived to be so useful in this research. The surveyed teachers expressed their unpleasant feelings about interacting with students online, such as a lack of visual or paralinguistic cues, and delayed and indirect feedback. The moderately low scores for all three activities pertaining to this role reflect the negative responses. These activities are sharing ideas on an e-learning wall, in-class bullet chat, and roll call to communicate with a certain student. Lecturers who use an e-learning wall demonstrate superior technology skills, so they are more likely to perceive ease of use and adopt RC for teaching. Bullet chat is a major function to implement discussion among teacher and students during a live broadcast lecture. Nonetheless, faculty members commented that the delayed text display and difficult attention towards personal students led to unsatisfactory communication, which might make lecturers perceive bad output quality. It should be noted that a random roll call for one-on-one communication in class, a valuable interaction approach in the traditional context, recorded a small negative value among online activities. The interviewees attributed the negative experiences to invisible body language and network delay when communicating with a student, but some regarded it as an essential task that they had to practice in online classes. Thus, the more teachers perform the activity, the less possibly they would adopt the technological system. There is another activity—uploading learning materials (chiefly PPTs), to present knowledge to learners—that obtained concordant negative values from three linear classifiers. This is the only activity contained in the role of instructional design and presenting content, as the statistics of other teachers' practices relevant to R1 are not offered by the RC system. In a face-to-face environment, teachers usually present all their knowledge in a conversational and attractive manner, yet on the RC platform there is no reminder or communication to motivate students to value these learning resources. As such, teachers formed negative perceptions through their comparisons with experiences in a face-to-face scenario. Generally, in a traditional class, lecturers can simultaneously receive non-verbal feedback (such as nodding heads) and verbal feedback (such as questions and

comments) from students [49]; in online teaching, instructor–student communication is primarily computer mediated, often involving asynchronous text-based exchanges, and thus lacks the physical nuances and immediacy of face-to-face interactions [50].

The findings of the current research are confirmed by some other studies. The rich literature has discussed the poor functionality of social interaction in an online environment. In the study [51], faculty members expressed the limited interaction with students in distance education where they are restricted by the demands of the camera and cannot observe participants via body language, verbal response, eye contact, etc. Chen et al. [52] claimed that on account of the space distance, interaction between teachers and students is difficult to achieve. Wang et al. [53] conducted a general evaluation of online learning for students from different dental schools during the COVID-19 outbreak, and found that interaction between teachers and students showed the lowest satisfaction. As for other teacher roles, Chou and Chou [3] investigated the continuance of online teaching and summarized that teachers were concerned with students' performance, the effectiveness of assessments, and attendance rates, as well as interaction difficulties with students. Gurley [54] examined exemplary educators' behaviors in blended and online learning environments, and reported that some behaviors (e.g., utilizing learning management system tools, course assignments, evaluation, and feedback) can facilitate teacher presence, whereas direct instruction is challenging. These findings are highly consistent with our results generated based on machine learning techniques. According to the useful roles and activities currently identified, we recognize and advocate that lecturers who teach online may transform their functions into assessing and facilitating learning from transferring knowledge in a traditional context.

As the cognitive instrumental process explained, system characteristics, interweaved with teaching goals, jointly affected lecturers' PU for their adoption of online teaching. Since a feature-related activity matches a corresponding function of the artifact, feature scoring can not only recommend useful roles and activities to teachers for their behavioral changes, but also provide artifact designers with cues and strategies on system adaption. Surry and Farquhar [55] (p. 3) called for "a new vision of instructional development in which the success of an instructional product is measured by its successful adoption just as much as success is measured by its instructional effectiveness". Hence, targeting sustainable online teaching, feature scores naturally group the corresponding system functions into three categories—with high positive scores, with low positive scores, and with negative scores. The respective strategies are listed as follows:

For functions with high positive scores, a recommender system could be integrated into the artifact to encourage lecturers' usage by presenting exemplary practices and pedagogical knowledge relevant to these functions;

For functions with low positive scores, four determinants of PU involving cognitive instrumental process should be analyzed for system devisers to judge whether these functions should be strengthened, eliminated, or replaced;

For functions with negative scores—especially high values—system devisers should respond to the urgent cues by overcoming the technological limitations to promote these functions, or by recommending, for example, a face-to-face approach instead of an online manner.

In order to complement the above strategies, devisors should conduct surveys to clarify users' perceptions and to capture the exact system weaknesses by questionnaires, interviews, or comments data.

## 6. Limitations

The limitations of the present study are as follows:

Firstly, the rankings of activities and teacher roles are merely the results of a relative comparison of PU, since feature scores have no definite threshold or baseline to identify usefulness or uselessness;

Secondly, the useful roles for online teaching are determined by both teachers' characteristics and system affordances, so, at different times, they would change along with the development of teachers' pedagogical competencies and system functionality;

Lastly, it should be noted that the current research is conducted at one university based on a particular artifact, although the research context is general in higher education in China.

## 7. Conclusions

This study presents a mechanism to support sustainable online teaching, hoping to compensate for the limited literature on scaffolding of teachers' professional competencies and technological systems' functionalities. Based on machine learning techniques, the mechanism can use lecturers' behavioral data automatically collected by the artifacts, in order to predict different types of teachers, identify useful teaching roles and activities, and inform both lecturers and system devisers about characteristics that they should change gradually to foster mutual adaption for the sustainable development of online teaching. Within a particular teaching system, classical classifiers forecasted individual teacher retention with up to 77% accuracy based on only three weeks' records, which made timely assistance feasible. According to feature scores, useful activities (e.g., in-class exercises, monitoring students' attendance, after-class formal testing, etc.) and teacher roles (such as learning assessment, guiding technology usage, and learning support) were determined to inform the focal lecturers who need to constantly change their functions to adapt to the given virtual settings. Although common views regarding exemplary roles for online teaching imply the necessity of teachers' transition from chiefly showing charisma in a face-to-face mode to e-moderators of learners, the lecturers' perceived useful roles and activities in the setting of a specific system depend both on their pedagogical beliefs and the artifact characteristics, which may be discordant with expert opinions. Since the useful activities that can foreshadow lecturer retention of online teaching will vary in dissimilar settings and are constantly changing over time, the presented mechanism should be embedded into artifacts to work regularly, nurturing coevolution between teachers' pedagogical competencies and technological system affordances.

**Author Contributions:** Conceptualization, Y.S., methodology, Y.S., formal analysis, Y.S., investigation, Y.S., resources, Y.S., data curation, Y.S., writing—original draft preparation, Y.S., writing—review and editing, Y.S.; software, F.G.; validation, Y.S. and F.G. All authors have read and agreed to the published version of the manuscript.

**Funding:** This research was funded by the Special Project for Teachers' Professional Development of Henan University, grant number YB-JFZX-13, and the National Natural Science Foundation of China, grant number 62107014.

**Institutional Review Board Statement:** Not applicable.

**Informed Consent Statement:** Not applicable.

**Data Availability Statement:** Restrictions apply to the availability of these data. Data were obtained from Henan University and are available from the authors with the permission of Henan University.

**Conflicts of Interest:** The authors declare no conflict of interest.

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
