# Peer review of "Exploring Useful Teacher Roles for Sustainable Online Teaching in Higher Education Based on Machine Learning"

_sustainability, doi:10.3390/su142114006_

Round 1
Reviewer 1 Report
This paper aims at exploring teacher roles for sustainable online teaching in higher education based on machine learning. However, I have some major concerns.
First, obvious English typos and formatting mistakes are everywhere on the paper.
Second, authors simple take advantage of tradtional machine learning methods for classification while the most advanced algorithms are not methioned. Besides, the experimental results is puzzling for me. For one thing, all the classifiers seem comparable to each other for their tasks, which is not the case for other domain. On the other hand, F1-score seems not correctly described(pls check your table 3. Performance results of applied classifiers in Track1 and Track2 )
Third, the recent three years' references are sufficient for this paper.
All in all, I cannot recommend the paper accepted by the journal.
Reviewer 2 Report
The authors discuss results of a study which explores activity patterns of university lecturers in an online teaching tool using machine learning algorithms. The topic investigated in this study might be significant for the readers of sustainability. However, there are some issues I would like to highlight.
-The objectives of the study are not always presented clearly and consistently in the manuscript. The introduction implies that the main goal is to predict online teaching in the future. Later in the text, an optimal set of roles should be found. First in line 232 it is mentioned that the study has two objectives, one precriptive and one descriptive. I recommend to state this to main goals of the study clearer and more early in the manuscript.
-Please expand the information on the feature Num_RedEnvelope: Is it really the case that teachers can transfer real money to students for their performance?
-Please provide some more specific information on how the data was anonymized? Were the lecturers informed and did they consent to the study?
-For me as a reader it remains unclear, what is meant by the term "optimal set of teacher roles". (e.g. line 168, 244, 628). Optimal according to which criteria? The results describe activities of the lecturers in the online teaching tool in the first three weeks of a semester, which are ‚associated‘ with an teaching approach with a certain proportion of e-learning. Are these activities optimal in the sense that teachers who have used them frequently before continue to use them later? After all, the optimal set of activities depends on what the goals of a course are, rather than whether or not it is online.
-The whole topic of eLearning is disscussed in a rather unreflective way in the manuscript. Although many universities want online courses established during the pandemic to continue afterward, that doesn't directly mean it's automatically beneficial or sustainable. As always in teaching, it depends on teaching goals which methods are useful. The manuscript would benefit if the authors discuss the benefits of eLearning more differentiated
-The overall argument regarding sustainable implementation of online teaching is not entirely convincing based on the results. Resp. the results are not discussed in a very differentiated way. Although the VIFs indicate significant features for the prediction of future online teaching approaches, the results can be interpreted similar to correlations. What comes first? Do lecturers' experiences with the tool lead them to prefer teaching online in the future? Or do teachers who like to teach online use the tool more extensively from the beginning? In general, other influencing factors should be included more in the discussion. For example, did the lectures know that after three weeks they will return to normal teaching in presence? If yes, this maybe influenced their behavior in the online tool, which then could influence the central features for prediction. Have teachers been considerate of student requests? Maybe students preferred analog teaching and lecturers followed their wishes/needs?
-The TAM as a theoretical model ist appropriate. However, the discussion could be improved by referring to models which take more influencing factors for intentions to conduct online teaching into account (like the Theory of planned behavior). This could enrich the intepretation of the results and the limitations of predicting future behavior solely based on activitities in the first three weeks.
- The survey of teachers is presented only very briefly (lines 288-290), but the survey results are used extensively in the discussion. It would be good if at least some more detail were provided on how and for what exactly the teachers were surveyed and how the survey was analyzed.
- Finally, I would like to ask a somewhat heretical question: the ML analyses were performed well and with a high level of effort. Looking at the results and the interpretation: would the same conclusions have been reached if the existing data had simply been used in a logistic regression model without ML? What is the further advantage of ML analysis?
Round 2
Reviewer 2 Report
The revision has substantially improved the clarity and coherence of the manuscript. All my comments have been adequately adressed. In particular, the discussion is now much more differentiated and the results are interpreted in more detail. Also, this makes the results more significant beyond the specific research context.